# The Outcome of Chemotherapy for Metastatic Extramammary Paget’s Disease

**DOI:** 10.3390/jcm10040739

**Published:** 2021-02-12

**Authors:** Hiroki Hashimoto, Yumiko Kaku-Ito, Masutaka Furue, Takamichi Ito

**Affiliations:** Department of Dermatology, Graduate School of Medical Sciences, Kyushu University, Fukuoka 812-8582, Japan; kyumiko@dermatol.med.kyushu-u.ac.jp (Y.K.-I.); furue@dermatol.med.kyushu-u.ac.jp (M.F.); takamiti@dermatol.med.kyushu-u.ac.jp (T.I.)

**Keywords:** extramammary Paget’s disease, metastatic extramammary Paget’s disease, chemotherapy, targeted therapy, survival analysis, docetaxel

## Abstract

The efficacy and survival impact of conventional chemotherapies for metastatic extramammary Paget’s disease (EMPD) have not been fully elucidated. This study examined the long-term outcome of chemotherapy for this indication. We conducted a retrospective review of 21 patients with distant metastatic EMPD (14 patients treated with chemotherapy and 7 patients treated without chemotherapy). The response rate of chemotherapy and patient survival were statistically analyzed. Among the 14 patients treated with chemotherapy, 12, 1, and 1 patient received docetaxel, paclitaxel, and low-dose 5-fluorouracil plus cisplatin, respectively, as the first-line treatment. The response rate was 50.0% (7/14), and the disease control rate was 64.3% (9/14). The median progression-free survival (PFS) and overall survival (OS) were 16.8 and 27.9 months, respectively. Multivariate analyses revealed that chemotherapy was a significant factor for prolonged PFS (hazard ratio (HR) 0.22, *p* = 0.038) but not for OS (HR = 1.71, *p* = 0.54). Ten patients (71.4%) had severe (grade 3 or 4) hematological adverse events. Although conventional chemotherapy improved PFS, we failed to show a significantly improved OS. Considering the frequent adverse events of conventional chemotherapy, targeted therapy may become a mainstay for the treatment of metastatic EMPD.

## 1. Introduction

Extramammary Paget’s disease (EMPD) is a rare neoplastic condition that was first described by Crocker in 1889 [1]. It commonly affects areas rich in apocrine sweat glands, including the vulva and perineal, perianal, scrotal, and penile skin of elderly individuals [2,3,4]. Most EMPD tumors remain restricted to the epidermis as in situ lesions, and they carry a good prognosis because of their slow-growing nature [2,5]. However, approximately 15–40% of EMPD lesions display dermal invasion, which increases the risks of lymph node and distant metastasis [3,6,7,8], resulting in a poor prognosis [5,9].

For unresectable distant metastases of EMPD, conventional chemotherapy has been used as the first-line treatment. Currently, no consensus has been attained regarding the optimal chemotherapy for the treatment of metastatic EMPD because of the rarity of the disease and the lack of clinical trials [10]. The effectiveness of several chemotherapeutic regimens was described in previous case reports or case series, but the numbers of patients involved were small. The described regimens included docetaxel (DTX) monotherapy [11,12,13]; low-dose 5-fluorouracil plus cisplatin (FP) [14,15]; 5-fluorouracil, epirubicin, carboplatin, vincristine, and mitomycin (FECOM) [16,17]; DTX plus S-1 (a drug containing a 5-FU derivative) [18,19,20]; S-1 monotherapy [21,22]; and cisplatin, epirubicin, and paclitaxel combination therapy [23,24]. However, the long-term efficacy and survival durations of these chemotherapies have not been fully elucidated. Recently, targeted therapies using monoclonal antibodies against human epidermal growth factor receptor 2 (HER2) have been applied for metastatic EMPD with some success [25,26,27]. Although these therapies shrink tumors in certain patient populations, no report compared patient survival in the absence of chemotherapy, and the effects of these agents on patient survival have not been fully evaluated.

In our institution, DTX monotherapy has often been used as the first-line chemotherapy for metastatic EMPD based on its weaker toxicity and certain tumor-shrinking effects. In this study, we described our experience in the management of patients with metastatic EMPD who were treated at our institution. The purpose of this study was to elucidate the impact of chemotherapy on long-term outcomes.

## 2. Materials and Methods

### 2.1. Data Collection

This retrospective review of our patients was conducted according to the guidelines of the Declaration of Helsinki. This study was approved by the Ethics Committee of Kyushu University Hospital (30–363; 27 November 2018). We retrieved the data of 220 patients with primary EMPD lesions. These patients were treated at the Department of Dermatology, Kyushu University (Fukuoka, Japan) between January 1997 and December 2020. At least three experienced dermatopathologists confirmed the diagnosis. Secondary EMPD, a direct expansion from a visceral organ cancer, was excluded. From these 220 patients, we collected the data of patients with distant metastasis (stage IV). Distant metastasis was identified via imaging (ultrasonography, chest X-ray, computed tomography (CT), and/or positron emission tomography with CT (PET/CT)). Lymph node metastasis beyond the regional lymphatic basin (e.g., superficial inguinal, deep inguinal, external iliac, and obturator lymph nodes) was considered as distant metastasis.

In total, 21 of 220 patients (9.5%) had distant metastasis. The following data on patients with distant metastasis were retrieved from our prospectively maintained databank and then analyzed: demographic data (sex, age at metastasis, and Eastern Cooperative Oncology Group (ECOG) performance status), clinical data (tumor site, clinical manifestation, and primary lesion size), and histopathological data obtained via hematoxylin and eosin staining (tumor thickness, which was measured to the second decimal place, as per the latest melanoma classification guidelines of the American Joint Committee on Cancer [28]). For patients with two or more primary lesions, we recorded the greatest tumor thickness and total tumor size. The distant metastatic sites and the number of metastases were also recorded. To count the number of distant metastases, lymph node metastasis was categorized by location as mediastinal, abdominal, or pelvic lymph node metastasis.

These patients were divided into two groups according to the receipt of chemotherapy/targeted therapy. The chemotherapy regimens used in our institution were as follows: DTX monotherapy consisted of tri-weekly or monthly administration of DTX at a dose of 60–75 mg/m^2^ (the actual dose was reduced at the physician’s discretion), DTX plus S-1 consisted of DTX at a dose of 40 mg/m^2^ (day 1) and S-1 at a dose of 40 mg/m^2^ (day 1–14) every 4 weeks; paclitaxel (PTX) monotherapy consisted of weekly PTX administration at a dose of 80 mg/m^2^; and low-dose FP therapy consisted of infusions of 5-fluorouracil at a dose of 500 mg/m^2^ and cisplatin at a dose of 5 mg/m^2^ five days/week. Only one patient was treated with DTX plus trastuzumab combination therapy, and the patient received DTX at a dose of 75 mg/m^2^ plus trastuzumab at a dose of 8 mg/kg as the first dose, followed by tri-weekly DTX at a dose of 75 mg/m^2^ plus trastuzumab at a dose of 6 mg/kg thereafter. The characteristics of patients were compared according to the receipt of chemotherapy. Patients who received chemotherapy were examined in detail regarding the types of anti-tumor drugs and the response to treatment. The size of metastatic lesions was evaluated using CT. The response to treatment was judged using the Response Evaluation Criteria in Solid Tumors as complete response (CR), partial response (PR), stable disease (SD), or progressive disease (PD). Adverse effects were evaluated using the National Cancer Institute’s Common Terminology Criteria for Adverse Events version 5.0 and scored from grade 1 to grade 4.

### 2.2. Follow-Up

The patients were monitored by physical examination every 3–6 months and imaging (ultrasonography, chest X-ray, and/or CT). Survival data, including the duration of survival and cause of death, were recorded.

### 2.3. Statistical Analysis

All statistical analyses were performed using JMP version 14.2 (SAS Institute, Cary, NC, USA). The χ^2^ test or Fisher’s exact test was used to analyze categorical variables, whereas the Mann–Whitney U test was used to analyze continuous variables. We used the Kaplan–Meier method to evaluate progression-free survival (PFS) and overall survival (OS), and we compared survival curves using the log-rank test. PFS was calculated from the date of diagnosis of distant metastasis to that of tumor progression diagnosed via imaging (CT and/or PET/CT), death, or the last follow-up prior to 31 December 2020. OS was calculated from the date of diagnosis of distant metastasis to that of death attributable to EMPD or the last follow-up prior to 31 December 2020. Data for patients who did not die were censored on 21 October 2020. Data for patients who died of other causes were censored at the time of death. The associations of clinical factors with PFS/OS were determined using a multivariate Cox proportional hazards regression model. Probability values less than 0.05 were regarded as statistically significant.

## 3. Results

### 3.1. Clinicopathological Data of the Patients

The demographic and clinical data of 21 patients with metastatic EMPD are presented in Table 1. All patients were Japanese, and their mean age was 74.8 years (range, 58–93 years). The cohort included 15 males (71.4%) and 6 females (28.6%). ECOG performance status was 0 or 1 in all patients. Tumors were predominantly located in the genital area (*n* = 17, 81.0%). Multiple lesions or tumors spreading over two areas were detected in three patients (14.3%). Nine patients (42.9%) had small primary lesions (<25 cm^2^), and 12 patients (57.1%) had large lesions (≥25 cm^2^). All primary lesions displayed dermal invasion. The tumor thickness was ≤4 mm in 9 patients (42.9%) and >4 mm in 12 patients (57.1%). Fifteen patients (71.4%) had distant lymph node metastasis beyond the regional lymphatic basin. Other metastatic sites included the liver (*n* = 10, 47.6%), lungs (*n* = 8, 38.1%), bone (*n* = 7, 33.3%), and brain (*n* = 3, 14.3%). Sixteen patients (76.2%) had distant visceral organ metastasis, whereas the remaining five patients (23.8%) had only distant lymph node metastasis. The number of distant metastatic sites was one in four patients (19.0%), two in eight patients (38.1%), three in five patients (23.8%), and four in four patients (19.0%).

### 3.2. Differences in Patient Characteristics According to the Receipt of Chemotherapy

The 21 patients were divided into two groups based on the receipt of chemotherapy/targeted therapy. Differences in patient characteristics between these groups are summarized in Table 2. Fourteen patients (66.7%) received conventional chemotherapy or a combination of chemotherapy and targeted therapy, and seven patients (33.3%) did not receive these treatments. No significant differences in background data were found between the two groups.

### 3.3. Treatment and Outcomes

The detailed data of 14 patients who received chemotherapy or targeted therapy are presented in Table 3. The median follow-up period was 18.0 months (range, 1.3–63.3 months). By the end of follow-up, three patients were alive, and 11 patients had died of EMPD. Ten patients (71.4%) were treated with DTX monotherapy as the first-line treatment. Other patients were initially started on PTX monotherapy (*n* = 1), low-dose FP (*n* = 1), or DTX plus S-1 (*n* = 1). Only one patient received targeted therapy (trastuzumab) in combination with DTX as the initial treatment. Radiation therapy combined with chemotherapy was provided to four patients with six lesions, and their metastatic sites were the pelvic lymph nodes (*n* = 3), brain (*n* = 2), and bone (*n* = 1). Of the 10 patients treated with DTX monotherapy, three patients were switched to low-dose FP (*n* = 2) or PTX (*n* = 1) because of tumor progression. Among all 14 patients treated with chemotherapy, the final response was PR in seven patients (50.0%), SD in two patients (14.3%), and PD in five patients (35.7%). The response rate (RR, CR + PR) and disease control rate (DCR, CR + PR + SD) of all 14 patients were 50.0 and 64.3%, respectively. The median PFS and OS of the 14 patients treated with chemotherapy were 16.8 and 27.9 months, respectively. In addition, among the 10 patients who received DTX monotherapy as the first-line treatment, the response was PR in five patients (50.0%), SD in two patients (20.0%), and PD in three patients (30.0%). The RR and DCR of DTX were 50.0 and 70.0%, respectively. The median PFS and OS of patients treated with DTX monotherapy as the first-line treatment were 6.2 and 27.9 months, respectively.

Of the seven patients who did not receive chemotherapy/targeted therapy, two patients underwent radiation therapy for brain (*n* = 1) and bone (*n* = 1) metastases. The median follow-up period was 8.9 months (range, 2.6–43.0 months). By the end of follow-up, two patients were alive, and five patients had died of EMPD. 

Patients treated with chemotherapy exhibited significantly longer PFS than those who did not receive chemotherapy (median PFS: 16.8 months vs. 3.1 months, *p* = 0.012). There was no significant difference in median OS between the two groups, although the survival tended to be prolonged in patients treated with chemotherapy (27.9 months vs. 11.9 months, *p* = 0.63). The Kaplan–Meier PFS and OS curves of patients with distant metastasis stratified by the receipt of chemotherapy are presented in Figure 1a,b.

The associations of the metastatic factors (the site, the number) and their survivals (PFS, OS) were also analyzed among the 14 patients treated with chemotherapy/targeted therapy. Table 4 shows the univariate analysis of patients treated with chemotherapy/targeted therapy for PFS and OS. The one-year PFS and OS of patients with distant visceral organ metastasis were 54.6% and 72.7%, which was not statistically significant compared with those without visceral organ metastasis (*p* = 0.24 and 0.45, respectively). In addition, the number of metastatic sites did not correlate significantly with PFS or OS (*p* = 0.35 and 0.31, respectively), although the survival rate tended to be lower when the number of metastatic sites was three or more.

### 3.4. Multivariate Analyses

The possible clinical factors associated with PFS were evaluated using multivariate Cox proportional hazards regression models. The following factors were included as explanatory variables: sex, age, tumor site, chemotherapy, and radiation therapy for metastatic sites. The results are listed in Table 5. Univariate analysis revealed that the receipt of chemotherapy was a statistically significant factor for prolonged PFS (hazard ratio (HR) = 0.27, *p* = 0.020).

Multivariate analysis results also confirmed that chemotherapy was a statistically independent factor associated with PFS (HR = 0.22, *p* = 0.038). However, chemotherapy did not significantly improve OS in multivariate analysis (HR = 1.71, *p* = 0.54). The results of multivariate analysis of OS in patients with distant metastasis are available in Appendix A.

### 3.5. Adverse Events of Chemotherapy and Targeted Therapys

Patients treated with DTX experienced high rates (8/12, 66.7%) of grade 3 or 4 myelosuppression. PTX monotherapy and low-dose FP also caused grade 3 myelosuppression. The injection of granulocyte-colony stimulating factor was required for patients with severe (grade 3 or 4) neutropenia. However, no treatment-related deaths occurred, and no patients had to discontinue chemotherapy because of adverse events. No adverse events were observed in patients treated with targeted therapy. A summary of treatment-related adverse events is given in Appendix A.

## 4. Discussion

Metastatic EMPD is an uncommon presentation of a rare cutaneous malignancy, and the standard treatment for metastatic EMPD has not been established. Therefore, this study evaluated the efficacy of conventional chemotherapy in terms of patient survival.

Chemotherapy has been the first-line treatment for unresectable distant metastases of EMPD. Recently, targeted therapies, such as trastuzumab and lapatinib, have been employed in patients with unresectable metastatic EMPD that was resistant to conventional chemotherapy [6,29,30]. However, no consensus regarding the optimal chemotherapy for EMPD has been reached because of its rarity and the lack of clinical trials. The effectiveness of several chemotherapeutic regimens has been described in case reports and small retrospective studies. Of the previously tested regimens, DTX monotherapy [11,12,13]; low-dose FP [14]; FECOM [17]; and the cisplatin, epirubicin, and paclitaxel [24] combination (given as first-line treatments) were associated with RRs of 31.8–58.3, 59.0, 57.1, and 80.0%, respectively; median PFS times of 6.0–9.0, 5.2, 6.5, and 8.0 months, respectively; and median OS times of 16.6–not reached, 12.0, 9.4, and 20.1 months, respectively. The studies of DTX monotherapy [11,12,13] had relatively large cohorts of patients (13, 14, and 22 patients, respectively), whereas the studies of other regimens were case reports or small studies with fewer than 10 patients [17,24].

In our institution, 12 out of 14 patients treated with chemotherapy received DTX therapy as the first-line therapy. Ten patients were treated with DTX monotherapy, and two patients were treated with DTX plus tegafur and trastuzumab. DTX monotherapy was linked to an RR of 50.0%, a median PFS of 6.2 months, and a median OS of 27.9 months, which were consistent with previous findings [11,12,13]. The other two patients were started on PTX monotherapy and low-dose FP therapy, respectively. The median PFS of all patients treated with chemotherapy was 16.8 months, which was significantly longer than that in patients who did not receive chemotherapy (*p* = 0.012). The median OS was 27.9 months. No statistically significant improvement of OS was observed (*p* = 0.63), although the survival tended to be prolonged in patients treated with chemotherapy. Multivariate analysis revealed that chemotherapy was a significant factor for prolonged PFS (HR = 0.22, *p* = 0.038) but not for OS (HR = 1.71, *p* = 0.54). These results suggest that chemotherapy is effective in terms of tumor control, although its efficacy may be temporary and prolonged OS cannot be expected.

Regarding safety, serious adverse events represent a major concern of conventional chemotherapy. Grade 3 or higher hematological adverse events were observed in all patients who received the DTX monthly regimen [11]. The cisplatin, epirubicin, and paclitaxel combination [24] was more effective than other regimens, including an RR of 80.0% and a median OS of 20.1 months. However, grade 3 or higher hematological adverse events were observed in 80.0% of patients, requiring adjustment of the dosing interval or the administration of granulocyte-colony stimulating factor. In our cohort, 71.4% of patients had grade 3 or higher hematological adverse events; therefore, it is necessary to establish a new regimen with fewer adverse events. Recently, a weekly low-dose DTX regimen was proposed [12], and low levels of hematological adverse events were expected to occur, although the RR might decrease.

Recently, targeted therapy (monotherapy or combination therapy with conventional chemotherapy) has been reported to be effective for some cases of metastatic EMPD [25,26,27,31,32,33,34]. Furthermore, a phase 2 study of trastuzumab plus docetaxel for HER2-positive metastatic EMPD is currently ongoing in Japan (UMIN000021311). In our case, the combination of DTX and trastuzumab was given to one patient as the first-line treatment, and the patient is being followed up after the observation of shrinkage of the metastatic lesions (liver and bone). However, because only one patient in our present study received targeted therapy, the impact of targeted therapy on survival may not be substantial. It is noteworthy that the effect of conventional chemotherapy alone on OS was limited in the current study.

This study was limited by the inherent potential bias of retrospective studies and the relatively small number of patients. The possibility still exists that the insufficient numbers of patients have led to the non-significant results in OS. In addition, only a few patients received regimens other than DTX monotherapy, and particularly, only one patient underwent targeted therapy. Further investigation with a larger cohort or a multi-center analysis will be necessary to support the current results.

## 5. Conclusions

In conclusion, we retrospectively reviewed the data of patients with distant metastasis of EMPD and assessed the benefit of chemotherapy. Although conventional chemotherapy improved PFS, we failed to show a significantly improved OS. Considering the frequent adverse events of conventional chemotherapy and the high tolerability of targeted therapy, targeted therapy may become a mainstay for the treatment of metastatic EMPD. Further investigation is needed to develop better treatment strategies for this rare disease.

## Figures and Tables

**Figure 1 jcm-10-00739-f001:**
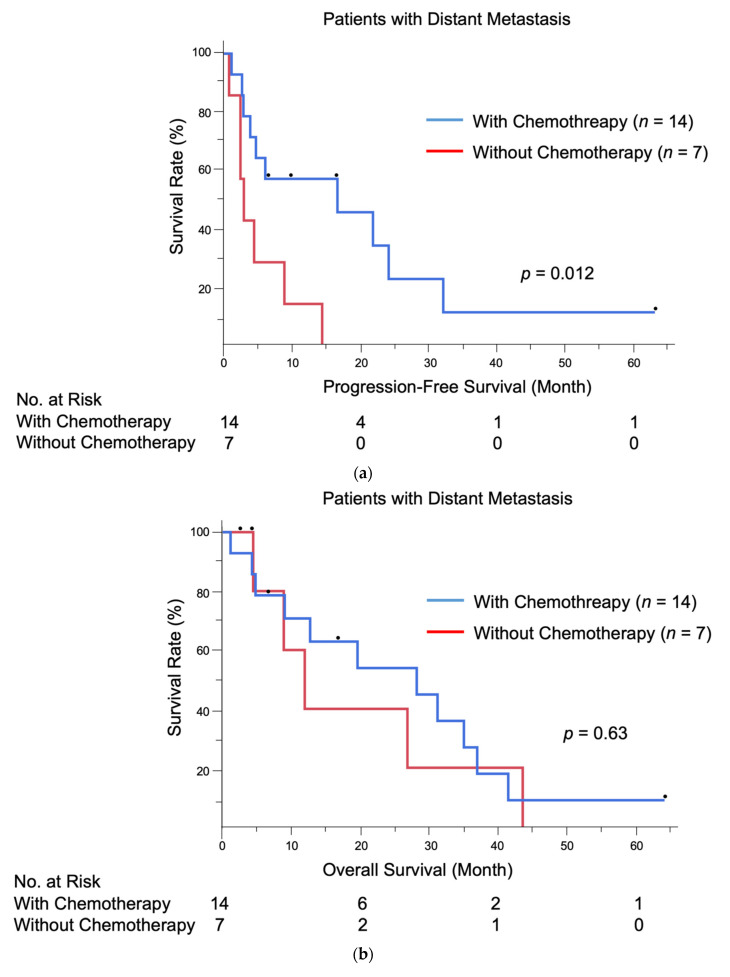
(**a**) Kaplan–Meier progression-free survival curves of 21 patients with distant metastasis; (**b**) Kaplan–Meier overall survival curves of 21 patients with distant metastasis.

**Table 1 jcm-10-00739-t001:** Demographic and clinical data of 21 patients with distant metastasis from extramammary Paget’s disease.

Parameter	*N* (%)
Sex	
Male	15 (71.4)
Female	6 (28.6)
Age at distant metastasis (years)	
Mean	74.8
Median (range)	75 (58–93)
ECOG performance status	
0	15 (71.4)
1	6 (28.6)
Tumor site	
Genital area only	17 (81.0)
Perianal area only	1 (4.8)
Genital + perianal areas	2 (9.5)
Genital + axillary + abdominal areas	1 (4.8)
Clinical manifestation	
Erythematous plaque	21 (100.0)
Nodule	17 (81.0)
Erosion/ulceration	13 (61.9)
Hypopigmentation	8 (38.1)
Primary lesion size (cm^2^)	
<25	9 (42.9)
≥25	12 (57.1)
TT (mm)	
In situ	0 (0.0)
≤4	9 (42.9)
>4	12 (57.1)
Site of distant metastasis	
Brain	3 (14.3)
Lung	8 (38.1)
Liver	10 (47.6)
Bone	7 (33.3)
Mediastinal LN	3 (14.3)
Abdominal LN	13 (61.9)
Pelvic LN ^†^	7 (33.3)
Distant visceral organ metastasis	
Present	16 (76.2)
Absent (only distant LN metastasis)	5 (23.8)
Number of metastatic sites	
1	4 (19.0)
2	8 (38.1)
3	5 (23.8)
4	4 (19.0)
Total	21 (100.0)

^†^ External iliac and obturator lymph nodes were excluded. ECOG, Eastern Cooperative Oncology Group; TT, tumor thickness; LN, lymph node.

**Table 2 jcm-10-00739-t002:** Demographic and clinical data of 21 patients with distant metastasis from extramammary Paget’s disease according to the receipt of chemotherapy.

Parameters	Chemotherapy/Targeted Therapy		*p* *
	Conducted (*n* = 14)	Not Conducted (*n* = 7)	
Sex			0.35
Male	11 (78.6%)	4 (57.1%)	
Female	3 (21.4%)	3 (42.9%)	
Age at distant metastasis (years)			0.057
Mean	71.7	80.9	
Median (range)	74 (58–82)	87 (68–93)	
ECOG performance status			0.12
0	12 (85.7%)	3 (42.9%)	
1	2 (14.3%)	4 (57.1%)	
Tumor site			0.26
Genital only	10 (71.4%)	7 (100.0%)	
Others	4 (28.6%)	0 (0.0%)	
Nodule formation			0.26
Present	10 (71.4%)	7 (100.0%)	
Absent	4 (28.6%)	0 (0.0%)	
Size of primary lesion			0.40
<25 cm^2^	5 (35.7%)	4 (57.1%)	
≥25 cm^2^	9 (64.3%)	3 (42.9%)	
TT (mm)			0.64
≤4	7 (50.0%)	2 (28.6%)	
>4	7 (50.0%)	5 (71.4%)	
Distant visceral organ metastasis			1.00
Present	11 (78.6%)	5 (71.4%)	
Absent (only distant LN metastasis)	3 (21.4%)	2 (28.6%)	
Brain metastasis			1.00
Present	2 (14.3%)	1 (14.3%)	
Absent	12 (85.7%)	6 (85.7%)	
Lung metastasis			0.17
Present	7 (50.0%)	1 (14.3%)	
Absent	7 (50.0%)	6 (85.7%)	
Liver metastasis			1.00
Present	7 (50.0%)	3 (42.9%)	
Absent	7 (50.0%)	4 (57.1%)	
Number of metastatic sites			0.81
1	2 (14.3%)	2 (28.6%)	
2	6 (42.9%)	2 (28.6%)	
3	3 (21.4%)	2 (28.6%)	
4	3 (21.4%)	1 (14.3%)	

Significant values are presented in boldface. * The Mann–Whitney U test was used for continuous variables, and the χ^2^ test or Fisher’s exact test was used for categorical variables. ECOG, Eastern Cooperative Oncology Group; CLND, complete lymph node dissection; TT, tumor thickness; LN, lymph node.

**Table 3 jcm-10-00739-t003:** Details of patients who received chemotherapy/targeted therapy for distant metastasis compared with patients without chemotherapy.

	First-Line Regimen						Without Chemotherapy (*n* = 7)
	Overall (*n* = 14)	DTX (*n* = 10)	DTX + Tegafur (*n* = 1)	DTX + Trastuzumab (*n* = 1)	PTX (*n* = 1)	Low-Dose FP (*n* = 1)	
Sex							
Male	11	8	0	1	1	1	4
Female	3	2	1	0	0	0	3
Age (years)							
Median (range)	74 (58–82)	76 (58–82)	76	73	61	65	87 (68–93)
Tumor site							
Genital only	10	7	0	1	1	1	7
Other sites	4	3	1	0	0	0	0
Metastatic site							
Brain	2	2	0	0	0	0	1
Lung	7	6	0	0	1	0	1
Liver	7	4	1	1	0	1	3
Bone	3	2	0	1	0	0	4
LN	14	10	1	1	1	1	7
Second-line regimen	Low-dose FP (*n* = 2) PTX (*n* = 1)	Low-dose FP (*n* = 2) PTX (*n* = 1)	None	None	None	None	-
Radiation therapy							
Done	4	4	0	0	0	0	2
Not done	10	6	1	1	1	1	5
Overall response ^†^							
CR	0	0	0	0	0	0	-
PR	7	5	0	1	0	1	-
SD	2	2	0	0	0	0	-
PD	5	3	1	0	1	0	-
FU, mo ^‡^							
Median (range)	18.0 (1.3–63.3)	23.4 (1.3–63.3)	12.7	16.6	4.4	41.0	8.9 (2.6–43.0)
Status at last FU							
Alive	3	2	0	1	0	0	2
DPD	11	8	1	0	1	1	5

^†^ The best response recorded from the start of chemotherapy until disease progression or death. ^‡^ The time between the date of diagnosis of metastasis and the last known follow-up. FU, follow-up; LN, lymph node; DTX, docetaxel; PTX, paclitaxel; FP, 5-fluorouracil + cisplatin; PR, partial response; SD, stable disease; PD, progressive disease; DPD, death from Paget’s disease.

**Table 4 jcm-10-00739-t004:** Univariate analysis of patients treated with chemotherapy/targeted therapy for progression-free survival and overall survival.

	Patients	One-Year PFS (%)	*p* *	One-Year OS (%)	*p* *
Sex					
Male	11	63.6	**0.049**	72.7	0.11
Female	3	33.3		66.7	
Age (years)					
<75	8	50.0	0.50	75.0	0.86
≥75	6	66.7		66.7	
Tumor site					
Genital only	10	60.0	0.23	70.0	0.19
Other sites	4	50.0		66.7	
Distant visceral organ metastasis					
Present	11	54.6	0.24	72.7	0.45
Absent (only distant LN metastasis)	3	33.3		66.7	
Lung metastasis					
Present	7	42.9	0.16	71.4	0.41
Absent	7	71.4		71.4	
Liver metastasis					
Present	7	57.1	0.54	71.4	0.44
Absent	7	57.1		68.6	
Number of metastatic sites					
1 or 2	8	62.5	0.35	75.0	0.31
≥3	6	50.0		62.5	

Significant values are presented in boldface. * Log-rank test. PFS, progression-free survival; OS, overall survival; LN, lymph node.

**Table 5 jcm-10-00739-t005:** Multivariate Cox proportional hazard analyses of progression-free survival among 21 patients with distant metastasis.

Variable	Univariate Analysis			Multivariate Analysis		
	HR	95% CI	*p*	HR	95% CI	*p*
Sex, male	0.38	0.13–1.12	0.079	0.66	0.16–2.75	0.56
Age (years) ^†^	1.01	0.95–1.07	0.79	0.99	0.93–1.06	0.84
Perianal lesion	0.78	0.22–2.82	0.71	1.96	0.28–13.76	0.50
Chemotherapy, conducted	0.27	0.089–0.81	**0.020**	0.22	0.052–0.92	**0.038**
Radiation therapy for metastatic sites, conducted	0.58	0.19–1.79	0.34	0.49	0.15–1.57	0.23

Significant values are presented in boldface. ^†^ Continuous variable. HR, hazard ratio; CI, confidence interval; LN, lymph node; CLND, completion lymph node dissection.

## Data Availability

The data presented in this study are available on request from the corresponding author. The data are not publicly available because of privacy restrictions.

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
