# Peer review of "The Outcome of Chemotherapy for Metastatic Extramammary Paget’s Disease"

_jcm, 2021, doi:10.3390/jcm10040739_

Round 1
Reviewer 1 Report
The Authors evaluate the outcome of chemotherapy for metastatic extramammary Paget's disease. The manuscript is interesting, however, it requires changes:
- The control group (patient not treated with chemotherapy) is small. Besides, two of these patients received radiotherapy. If the rest did not receive the treatment, these two patients should be excluded from the statistical analysis.
- The study group is very diverse, it would be interesting to divide patients into subgroups according to the type of chemotherapy they received.
- Figure 1 is illegible, authors should include a better quality figure
Author Response
Response to Reviewer 1 Comments
Revised Manuscript: “The outcome of chemotherapy for metastatic extramammary Paget’s disease," by Hashimoto et al.
Thank you very much for your constructive comments about our manuscript. We have addressed the comments as shown below.
The Authors evaluate the outcome of chemotherapy for metastatic extramammary Paget's disease. The manuscript is interesting, however, it requires changes:
- The control group (patient not treated with chemotherapy) is small. Besides, two of these patients received radiotherapy. If the rest did not receive the treatment, these two patients should be excluded from the statistical analysis.
→ Thank you for the comment. As the reviewer pointed out, the control group was small (seven patients), but despite the small number of patients, both univariate and multivariate analyses showed prolonged progression-free survival in patients treated with chemotherapy. Two of the patients in the control group received radiation therapy, but some of the patients treated with chemotherapy also underwent radiation therapy (4/10, 40%). Since the multivariate analysis showed that radiation therapy was not an independent prognostic factor for either progression-free survival or overall survival, and when the patients were excluded, the survival curves were similar to the present ones (median PFS: 16.8 months vs. 3.1 months, p = 0.033; median OS: 27.9 months vs. 11.9 months, p = 0.84), we would like to include these patients who received radiation therapy in the statistical analysis.
- The study group is very diverse, it would be interesting to divide patients into subgroups according to the type of chemotherapy they received.
→ Thank you for the comment. We agree with the comment that subgroup analyses would be interesting. According to the reviewer’s comment, we tried to perform subgroup analyses but the numbers of patients who received non-DTX therapy were too small to draw a meaningful result. We discussed on this issue in the limitation section in the revised manuscript. It is necessary to accumulate data of metastatic extramammary Paget' s disease for further study.
- Figure 1 is illegible, authors should include a better quality figure.
→ As the reviewer pointed out, we improved the resolution of figures and enlarged the size of figures. The text in the figure was also enlarged.
Finally, we truly appreciate the reviewer’s careful and constructive comments about our manuscript.

Reviewer 2 Report
The manuscript titled “The outcome of chemotherapy for metastatic extramammary 2 Paget’s disease” is interesting. The limitation of this study is that only one patient received targeted therapy and drawing the conclusion based on this might not be relevant. It has already shown in several studies the limitation of chemotherapy.
Comments to authors:
- Figure 1 needs better resolution and clarity.
- Are the authors considering other patients with Paget’s disease in their future studies?If not author can provide a limitation paragraph to justify the conclusion based on one patient data?
- Did the author study any other PK parameters?
Author Response
Response to Reviewer 2 Comments
Revised Manuscript: “The outcome of chemotherapy for metastatic extramammary Paget’s disease," by Hashimoto et al.
Thank you very much for your constructive comments about our manuscript. We have addressed the comments as shown below.
The manuscript titled “The outcome of chemotherapy for metastatic extramammary 2 Paget’s disease” is interesting. The limitation of this study is that only one patient received targeted therapy and drawing the conclusion based on this might not be relevant. It has already shown in several studies the limitation of chemotherapy.
Comments to authors:
- Figure 1 needs better resolution and clarity.
→ As the reviewer pointed out, we improved the resolution of figures and enlarged the size of figures and their text.
- Are the authors considering other patients with Paget’s disease in their future studies? If not author can provide a limitation paragraph to justify the conclusion based on one patient data?
→ Thank you for the comment. As the reviewer pointed out, only a few patients were used regimens other than DTX monotherapy, and particularly, only one patient underwent targeted therapy. We made a discussion in the limitation paragraph in the revised manuscript.
- Did the author study any other PK parameters?
→ Thank you for the comment. As the reviewer suggested, it would be interesting to evaluate the relationship between the blood concentration of anti-tumor drugs and their effect. However, we basically did not monitor the blood concentration of the anti-tumor drugs, and the dosage was not completely constant between patients because the actual dose was reduced at the physician’s discretion. Therefore, we did not study other PK parameters.
Finally, we truly appreciate the reviewer’s careful and constructive comments about our manuscript.

Reviewer 3 Report
Thank you for the opportunity to review the manuscript "the outcome of chemotherapy for metastatic extramammary Paget' s disease, this is a well written manuscript. The Quality of presentation is very good but nowadays target therapy should be analyzed to improve the clinical outcome
Author Response
Response to Reviewer 3 Comments
Revised Manuscript: “The outcome of chemotherapy for metastatic extramammary Paget’s disease," by Hashimoto et al.
Thank you very much for your constructive comments about our manuscript. We have addressed the comments as shown below.
Thank you for the opportunity to review the manuscript "the outcome of chemotherapy for metastatic extramammary Paget' s disease, this is a well written manuscript. The Quality of presentation is very good but nowadays target therapy should be analyzed to improve the clinical outcome.
→ Thank you for the comment. We agree with the comment. Targeted therapy should be analyzed to improve the clinical outcome, but in our present study, only one patient treated with targeted therapy was included, therefore we could not fully analyze the correlation between targeted therapy and their survival. It is necessary to accumulate data of metastatic extramammary Paget' s disease for further study. We truly appreciate the reviewer’s constructive comments about our manuscript.

Reviewer 4 Report
This is an interesting study, and results are important in refining the best treatments for this rare condition. The fact that it affects older adults, and that it is an uncommon and rarely metastatic disease, results in a relatively small sample size, which of course limits the robustness of the data and its interpretation, especially when there is a relevant degree of variation between treatments (doses, frequencies and combinations).
One of the main conclusions presented is that there is no significant difference seen in OS, which is followed by the median of each group (27.9 vs 11.9 months). This is rather striking, and even though it is not statistically significant would the authors comment on dismissing this rather pronounced trend?
Is it a result of an extreme outlier on either group? According to figure 1b, after 20 months overall survival was >50% for the chemotherapy treated group, and <30% for the non-treated group. In spite of the important statistical considerations, this appears of actual relevance to a reader and I believe needs to be acknowledged.
Tables could be more visually intuitive, as they are very "data-dense". For example, table 3 could be organised by treatment (for example, just DXT) instead of "case number". Colour or shade by treatment, or metastatic site (just one or multiple, for example) would make it more intuitive to cluster and identify patterns.
Also, is there any relation between the metastatic site or number or metastatic organs and the outcome or response rate?
All of the above would make the reading and understanding a little easier, and therefore transposing to the implications (or suggestions?) to future clinical refinements .
The data is as solid as it can be with such a small sample size, but it would be interesting to know how the authors see these results built on? More patients? Other databases? Comparisons across other institutions/hospitals/countries standard of care? The discussion section of the report could be deeper and broader to incorporate the implications of these results.
Author Response
Response to Reviewer 4 Comments
Revised Manuscript: “The outcome of chemotherapy for metastatic extramammary Paget’s disease," by Hashimoto et al.
Thank you very much for your constructive comments about our manuscript. We have addressed the comments as shown below.
- This is an interesting study, and results are important in refining the best treatments for this rare condition. The fact that it affects older adults, and that it is an uncommon and rarely metastatic disease, results in a relatively small sample size, which of course limits the robustness of the data and its interpretation, especially when there is a relevant degree of variation between treatments(doses, frequencies and combinations).
→ Thank you for the comment. We agree with the reviewer’s comment. Because of the rarity of metastatic extramammary Paget’s disease and the lack of clinical trials, only relatively small retrospective studies have been reported. This study also did not collect a sufficient number of patients, therefore the results obtained from the statistical analysis are limited. We made a discussion in the limitation paragraph in the revised manuscript.
- One of the main conclusions presented is that there is no significant difference seen in OS, which is followed by the median of each group (27.9 vs 11.9 months). This is rather striking, and even though it is not statistically significant would the authors comment on dismissing this rather pronounced trend?
→ As the reviewer pointed out, the overall survival seemed to be prolonged in patients treated with chemotherapy, although there was no significant difference in overall survival between the two groups (27.9 months vs. 11.9 months, p = 0.63). We added this information to the result in the revised manuscript. However, multivariate analysis failed to show the improvement of overall survival with chemotherapy. We failed to show a significantly improved OS.
- Is it a result of an extreme outlier on either group? According to figure 1b, after 20 months overall survival was >50% for the chemotherapy treated group, and <30% for the non-treated group. In spite of the important statistical considerations, this appears of actual relevance to a reader and I believe needs to be acknowledged.
→ Thank you for the comment. As shown in Table 2, there was no statistically significant difference in patient background between the chemotherapy-treated and non-treated groups. As the reviewer suggested, there was a discrepancy in overall survival depending on the month of follow-up. It is possible that this is due to the small cohort size of this study, and that therefore the difference was not statistically significant. We discussed on this issue in the limitation section of the revised manuscript. However, we could not prove the effect of chemotherapy on prolonged overall survival in our analyses. Therefore, we could not conclude that chemotherapy is effective in prolonging overall survival.
- Tables could be more visually intuitive, as they are very "data-dense". For example, table 3 could be organised by treatment(for example, just DTX) instead of "case number". Colour or shade by treatment, or metastatic site (just one or multiple, for example) would make it more intuitive to cluster and identify patterns.
→ Thank you for the comment. In accordance with the reviewer’s comment, we revised Table 3, which is organized by the treatment.
- Also, is there any relation between the metastatic site or number or metastatic organs and the outcome or response rate?
→ Thank you for the comment. As the reviewer pointed out, we conducted the additional statistical analyses among the 14 patients treated with chemotherapy, and the analyses showed no significant correlation between the site of metastatic organs or the number of metastatic sites and the survival rate (progression-free survival, overall survival). We added this result as Table 4 in the revised manuscript.
Regarding the relationship with the response rate, it was difficult to calculate the response rate by focusing on each metastatic site in this study because the effect of chemotherapy is not uniform for each metastatic site when there are multiple metastatic sites.
- All of the above would make the reading and understanding a little easier, and therefore transposing to the implications (or suggestions?) to future clinical refinements.
→ Thank you for the constructive comment. We are most grateful that the reviewer's comments improved our manuscript.
- The data is as solid as it can be with such a small sample size, but it would be interesting to know how the authors see these results built on? More patients? Other databases? Comparisons across other institutions/hospitals/countries standard of care? The discussion section of the report could be deeper and broader to incorporate the implications of these results.
→ Thank you for the comment. As the reviewer suggested, our study was limited by the inherent potential bias of retrospective studies, and the limited number of patients. To support the results of our study, it may be necessary to expand the study to multi-centers and accumulate the number of patients in future studies. We made a discussion in the limitation paragraph in the revised manuscript.
Finally, we truly appreciate the reviewer’s careful and constructive comments about our manuscript.

Round 2
Reviewer 1 Report
I would like to thank the Authors for their response to my comments.